# Critical Analysis of Non-Thermal Plasma-Driven Modulation of Immune Cells from Clinical Perspective

**DOI:** 10.3390/ijms21176226

**Published:** 2020-08-28

**Authors:** Barbora Smolková, Adam Frtús, Mariia Uzhytchak, Mariia Lunova, Šárka Kubinová, Alexandr Dejneka, Oleg Lunov

**Affiliations:** 1Department of Optical and Biophysical Systems, Institute of Physics of the Czech Academy of Sciences, 18221 Prague, Czech Republic; smolkova@fzu.cz (B.S.); frtus@fzu.cz (A.F.); uzhytchak@fzu.cz (M.U.); mariialunova@googlemail.com (M.L.); sarka.kubinova@iem.cas.cz (Š.K.); dejneka@fzu.cz (A.D.); 2Institute for Clinical & Experimental Medicine (IKEM), 14021 Prague, Czech Republic; 3Department of Biomaterials and Biophysical Methods, Institute of Experimental Medicine of the Czech Academy of Sciences, 14220 Prague, Czech Republic

**Keywords:** non-thermal plasma, cytotoxicity, cell signaling, immunomodulation

## Abstract

The emerged field of non-thermal plasma (NTP) shows great potential in the alteration of cell redox status, which can be utilized as a promising therapeutic implication. In recent years, the NTP field considerably progresses in the modulation of immune cell function leading to promising in vivo results. In fact, understanding the underlying cellular mechanisms triggered by NTP remains incomplete. In order to boost the field closer to real-life clinical applications, there is a need for a critical overview of the current state-of-the-art. In this review, we conduct a critical analysis of the NTP-triggered modulation of immune cells. Importantly, we analyze pitfalls in the field and identify persisting challenges. We show that the identification of misconceptions opens a door to the development of a research strategy to overcome these limitations. Finally, we propose the idea that solving problems highlighted in this review will accelerate the clinical translation of NTP-based treatments.

## 1. Introduction

Implementation of technologies from physics in medical practice has a long history [1] and still shows significant renovation and progress [2]. As a result, there are expectations that physics-based techniques will bring novel diagnostics and treatment modalities in post-genomic personalized medicine [1]. In the last two decades, the research has shown that non-thermal plasma (also known as cold atmospheric plasma or non-equilibrium atmospheric pressure plasma) has significant potential in various biomedical applications [3,4,5]. Indeed, non-thermal plasma (NTP) has emerged as a promising tool for the sterilization of medical equipment [6,7,8], wound healing [9,10], bacteria eradication [11,12,13,14], dental hygiene [6,15], blood coagulation [16], angiogenesis suppression [17], cancer treatment [4,18,19], and food decontamination [20,21]. It is worth noting here that, while some studies show angiogenesis suppression [17], others describe accelerated angiogenesis by NTP [10,22].

Among those various biomedical applications of NTP, bacteria eradication and wound healing showed significant progress toward clinical application [23,24]. Indeed, several clinical trials revealed the potential applicability of NTP to decrease the bacterial load on chronic wounds in patients [25,26,27,28]. This success resulted in the commercialization of a number of plasma sources for wound healing and skin treatment [29,30]. Finally, in 2013, some NTP sources got CE certification as medical devices for the treatment of chronic wounds and pathogen-based skin diseases [23,24]. However, the research on NTP sources continues to develop extensive characterization and optimization of plasma systems and their biological effects [30,31]. Studies related to novel biomedical applications of NTP are currently actively expanding [30]. Thus, it is expected that, in the near future, the NTP field will bring other interesting possibilities for biology and medicine [30,32].

Basically, NTPs represent ionized gases with ion temperatures close to room temperature [33]. The composition of an NTP is very complex and consists of ions, charged molecules, electrons, electric fields, free radicals, low amounts of UV radiation, and neutral molecules [33]. Emerging evidence suggests that among other components, reactive oxygen species (ROS) and reactive nitrogen species (RNS) act as key mediators of biological responses triggered by NTP treatment [34,35,36,37]. In fact, ROS and RNS are well-known to regulate and influence key cellular processes, such as cell growth, migration, proliferation, differentiation, death, aging, inflammation and regeneration [38,39,40,41,42,43,44]. Different studies repeatedly showed that NTP is able to generate a number of ROS, such as O, ^•^OH, O_2_^•−^, ^1^O_2_, NO^•^, NO_2_^•^, H_2_O_2_, NO_2_^−^, NO_3_^−^, or O_3_ [45]. Therefore, it is not surprising that NTP has such various biological effects. It is worth noting here that ROS and RNS intracellular actions also include promotion or suppression of inflammation, immunity, and carcinogenesis [44]. Thus, recent studies of the plasma field have focused on the potential immunomodulatory effects of NTP [46,47,48]. Research of such NTP applications is relatively new, but there are already substantial efforts and promising results toward immunomodulation by NTP. Therefore, this review aims to critically revisit the current literature about the NTP effects exerted on immune cells. One of the core principles of the scientific method is critical analysis. When we neglect critical assessment of the scientific literature, it leads to that questionable and irreproducible studies more likely stay unnoticed [49]. Irreproducible research not only wastes resources, hampers progress, and leads to frustration in academic science [50], but it also has devastating economical and personal consequences [51]. Thus, we herein describe and identify gaps in our understanding of underlying cellular mechanisms triggered by NTP. Identification of gaps in the scientific knowledge helps to shape the research process and provides a basement for the design of reliable and reproducible technologies. It is crucial to know the cellular mechanism of the treatment’s action. Lack of such knowledge may result in severe clinical failure of desired treatment modality [52,53]. Indeed, identification of molecular targets of NTP action will enable better clinical transition of the technology. Finally, we discuss current challenges and perspectives in the biomedical applications of NTPs.

## 2. A Brief Physicochemical Characterization of NTP for Biomedicine

In this section, we would like to give only a brief overview with key points on the complicated question of the physicochemical characterization of NTP. There are many high-quality and comprehensive reviews on this issue [24,30,33,45,54,55]. Here, we give only a short description that is necessary to further understand the cellular effects of NTPs.

Current progress in the design and fabrication of various NTP systems allows us to relatively easy generate plasmas with temperature under 40 °C using different gases, e.g., helium, argon, nitrogen, ambient air, or a mixture of gases [55,56]. One can divide NTP systems into two generalized categories based on device principles, i.e., dielectric barrier discharge (DBD) and atmospheric pressure plasma jet (APPJ) [29,33]. DBD generates plasma in the gap between an isolated (dielectric) high-voltage electrode and a biological specimen (serving as a counter electrode) being exposed to NTP [33,57]. A non-thermal discharge is formed between these two electrodes when an alternating current (AC) high voltage of varying kV and up to MHz is applied [33,45,57]. In fact, the majority of DBD devices utilize atmospheric air as working gas [33,45,57]. One disadvantage of DBD systems is the high ignition voltage (10 kV or more depending on the system’s configuration). Thus, certain precautions or isolations are essential in DBD devices [33,45,57]. Due to the direct contact with tissue, DBD devices have shown a very promising effect in blood coagulation and tissue sterilization [16]. It is possible to replace one of the electrodes with an object with high charge storage capacity, creating a so-called “floating electrode” (FE) [58]. One can utilize living tissue for charge storage [58]. The system is called a “floating electrode DBD” (FE-DBD) [58]. The advantage of the FE-DBD system is in the absence of thermal or chemical damages during application on living tissues [58,59]. However, one should not forget that DBD systems operate with a relatively high current that has to pass through living tissue. Therefore, caution should be taken into account not to exceed safety limits [33,45,57].

Contrary to DBD, APPJ sources do not use the target area (namely biological object) as a counter electrode [24,30,33,45,54,55]. Instead, two electrodes within a device are utilized to create NTP, and then the NTP is transported to a desired biological object via diffusion or by a carrier gas. Different carrier gases can be used to create NTP. This allows the plasma-generated reactive species to be modified relatively easily. Thus, one can tune the chemical composition of the resulting NTP to reach the desired plasma compositions for specific biomedical applications [24,30,33,45,54,55]. Indeed, APPJ-based devices vary greatly in design and performance. One can produce systems that range from thin plasma needles, jets, and multiple jet applications up to large-size plasma torches [24,30,33,45,54,55].

The aforementioned complexity of NTP composition leads to the long list of up to 96 chemical reactions taking place in air NTP [60]. This illustrates how complex the entire NTP composition is (Figure 1). In fact, the quality and quantity of chemical entities that form NTP greatly vary depending on the type of gas being used to generate plasma [24,30,33,45,54,55,60,61,62]. Additionally, there is a number of ROS and RNS being produced as a result of plasma–liquid interactions (Table 1) [24,30,33,45,54,55,60,61,62]. The actual state of knowledge indicates that the biological effects of NTPs are mediated to a large extent by ROS, RNS generated in NTP and/or transferred into irradiated tissue [24,30,33,45,54,55,60,61,62].

One has to bear in mind that those ROS/RNS generated by NTP can be sub-divided in short-lived with a half-life in the μs range (O, ^•^OH, O_2_^−^, ^1^O_2_, NO^•^, NO_2_^•^) and relatively long-lived with a half-life in the ms range (H_2_O_2_, NO_2_^−^, NO_3_^−^, O_3_) [63,64]. Additionally, one of the most abundant component of NTP (O_3_) has been shown to generate reactive oxygen intermediates (ROIs) with the chemical lifetime exceeding 100 s [65]. Thus, it is feasible that NTP would modulate redox reactions in living tissues. Of note, the penetration depth of plasma in tissues is relatively low [36,63,66]. It has been shown that NTP can reach a depth of 2 mm in vitro, contrary in in vivo conditions NTP penetrates only up to ~400 μm [36,63,66]. It was proposed that the oxidizing nature of NTP may trigger alterations in redox-sensitive reactions and in a way affecting the microenvironment in deeper layers of the irradiated tissues [34,36,67,68].

Indeed, ROS and RNS regulate the plethora of cellular processes [38,39,40,41,42,43,44]. As a result, NTP has been implicated to modify very distinct biological processes, ranging from increased proliferation [69,70] to cell death by necrosis [71] or apoptosis [72,73,74]. Furthermore, the biological effects of NTPs greatly depend on the physical and chemical characteristics of plasma used for the treatment [62,75,76,77]. Another degree of variability in NTP devices is a very big range (from 0.5 kV up to 100 kV) of the voltage producing discharges [33]. Furthermore, voltage frequencies that are used in different NTP sources vary enormously [33]. To sum up, different NTP sources vary in following major parameters of the system: feed gas compositions (e.g., N_2_, O_2_, artificial air (80% N_2_ + 20% O_2_), ambient air, Ar, He, mixture of gases), input power, discharge voltage, gas flow rate, jet length, voltage frequencies [24,30,33,45,54,55]. All these parameters affect the resulted physicochemical composition of NTP [24,30,33,45,54,55].

## 3. Critical Clinical View on NTP—Potential Side Effects and Clinical Validation

Another very important topic that we want to touch before going to effects on immune cells is potential side effects. Indeed, we feel that this is a very important topic that is not comprehensively covered in plasma literature. There are only a few studies that mention the side effects of NTPs [78,79,80]. Generally, adverse drug reactions or side effects occur almost daily in healthcare institutions and have to be carefully considered in the optimization of treatment modalities [81,82,83]. It is plausible that such a complex composition of NTP may trigger different effects including side effects on human cells. For instance, it has been shown that ROS and RNS, as well as radiation energy of NTP, may trigger cellular toxicity and cause DNA damage [84,85]. Moreover, in redox biology, it is well-known, that intracellular accumulation of excessive levels of ROS damages cellular structures, leading to distinct types of cell death [86,87,88,89]. In fact, it has been shown that NTP may induce focal mucosal erosion with superficial ulceration and necrosis accompanied by a mild inflammatory reaction [79]. We summarized current studies that assessed side effects of NTP in Table 2. We performed this analysis to illustrate that the question about side effects elicited by NTP is still open. As one can see from Table 2, there is very limited number of studies that address side effects of NTP. In fact, the parameters of NTP treatment (duration, NTP voltage, frequency of treatment, etc.) in different studies varied, which precluded direct comparison and analysis. However, our intention was to highlight the necessity and importance of such studies. Generally, the assessment of side effects of a treatment is very important because side effects are crucial parameters in successful clinical performance [90]. Additionally, side effects threaten patient compliance [90].

One can see that available studies indicate that NTP treatment has no severe but only mild side effects and is well-tolerated (Table 2). Indeed, wounds and skin diseases represent the majority of pathological conditions where the side effects of NTP were assessed (Table 2). However, NTP treatment endeavors to attain different clinical applications (not only wound healing) [4,18,19]. Therefore, studies on potential side effects and those related to treatment complications should be scaled up.

As we will see further, NTP-induced immune cell modulation has significant potential to be a new treatment modality for cancer pathologies [46,96]. This is very appealing and encouraging. However, before we come close to NTP-induced immunomodulation, we need to at least very briefly discuss current views on clinical verification of treatment efficacy and good clinical practice. In fact, bias can explain extraordinary results that were not confirmed further in many individual studies [97]. The study design of meta-analysis and randomized clinical trials helps to avoid bias in questions of treatment effectiveness [97]. As a result, importance and influence of evidence-based research in medicine is constantly growing worldwide [97,98]. Indeed, evidence-based research becomes one of the most crucial medical milestones that affects the development of clinical guidelines [97,98,99]. It is worth noting here that clinical practice guidelines (CPGs) now represent concrete practice recommendations for healthcare providers [100,101,102,103,104]. CPGs represent a component of evidence-based medicine [100,101,102,103,104]. According to the principles of evidence-based medicine, CPGs are based on the extensive evaluations of whether evidence likely supports the efficacy of the treatment when taking into account risk-of-bias concepts [100,101,102,103,104]. This evaluation led to formulation of the so-called “evidence pyramid” (Figure 2) [105,106,107,108].

In fact, each ascending level of the pyramid (Figure 2) is represented by improved quality of evidence and decreased risk of bias [105,106,107,108]. Meta-analysis performs a crucial role in the formulation of CPGs [109,110,111,112]. Careful analysis of studies dealing with NTP side effects clearly shows that the majority of the reports are either case studies or case-control studies (Table 2). There are only a few randomized controlled trials with no meta-analysis performed (Table 2). It is important to realize the necessity of such analysis. Lack of systematic summaries leads to extreme inconsistency between evidence and expert recommendations [97,98,99]. If such recommendations rely on low-quality evidence from individual studies and/or preconceptions, this leads to destructive and disastrous consequences in clinical practice [97,98,99]. Thus, we have to be careful in our justifications of the efficacy of NTP treatment and the absence of side effects.

## 4. Effects of NTP on Immune Cells

In the above section, we summarized that NTP has a very complicated physicochemical composition. However, ROS/RNS are now emphasized as major biological players of plasma. It is, indeed, plausible that ROS/RNS produced in plasmas are not necessarily the species directly affecting cells [113]. It is very likely that secondary products of oxidation and ROS/RNS formed and/or accumulated in cells play a greater role as biological effectors of NTP [36]. Taking into account that ROS and RNS participate, regulate, and modulate activity and responses of immune cells [44], NTP researchers have undertaken studies showing that NTP may also possess immunomodulation consequences [48].

It is worth noting here that a major function of the human immune system is to protect the body from infectious agents by different effector cells and proteins. In general, we recognize an innate immune system and an adaptive immune system. The innate immune system, consisting of phagocytic cells and natural barriers, is involved in unspecific host defense [114]. On the other hand, the adaptive immune system includes the antibody production and development of immunological memory in response to pathogens [115]. The progenitor hematopoietic stem cell may develop in myeloid (granulocytes, monocytes) or lymphoid (T, B, and natural killer cells) cells [115]. Monocytes, as effectors of the innate immune system, are responsible for engulfing pathogens and cellular debris in the human body [116]. In addition, monocytes are able to produce ROS and RNS, particularly nitric oxide radical (NO^•^) and hydrogen peroxide H_2_O_2_, to destroy phagocytized bacteria [117]. Upon infection, signal monocytes are recruited from the bloodstream to the place of inflammation and are differentiated into M1 or M2 macrophages. The process of macrophage polarization is driven by microenvironment at the site of inflammation [118].

It becomes evident that not only biochemical stimuli but also physical agents can stimulate the polarization of the macrophage and modulate the immune system [119,120]. Indeed, NTP treatment has shown the potential to enhance macrophage activation and polarization in vitro [121,122,123]. In fact, the literature on NTP effects on various immune cells is rapidly growing. Therefore, we propose here a critical overview of this topic. First of all, we summarized current state-of-the-art studies on NTP affecting immune cells in vitro in Table 3.

Indeed, there are more publications showing NTP effects on immune cells. However, we selected those that have shown a verifiable biological effect supported by rigorous methodology. We will come back to this point later, when discussing challenges with deciphering molecular mechanisms of NTP action. From Table 3, one can clearly see that NTP has potential in modulating immune cell activity with outcomes ranging from immune cells activation to induction of different kinds of cell death. A closer look at the results presented in Table 3 reveals that many studies used NTP treatment of immune cells not just for cellular function modulation solely but rather as a potential modality for cancer immunotherapy. In fact, recent advances in cancer immunotherapy showed that significant improvement in patient survival is possible with modern immunotherapy treatments [135,136]. Thus, the development of novel treatments (even maybe additive or complementary) in combination with existing immunotherapies is of a great importance. NTP may play here an emerging role to potentially improve clinical outcomes by supporting immunomodulatory effects.

THP-1 (acute monocytic leukemia cell line) showed greater resistance to plasma treatment in comparison to primary monocytes [137] and Jurkat (acute T cell leukemia cell line) cells [125,130]. NTP-treatment led to p53 [131] and caspase3/7 [130] activation and apoptosis execution in Jurkat cells. On the other hand, Kaushik et al. revealed that NTP led to the mitochondria membrane depolarization; cytochrome c release; and induction of apoptosis in THP-1, U937, and RAW264.7 cells [126]. Furthermore, NTP-treatment triggered neutrophil extracellular traps (NET) formation and the IL-8 release, perhaps as an outcome of cell death [127].

Further analysis of the Table 3 brings us to the conclusion that the majority of the research is done utilizing monocytic cell lineages. Furthermore, studies summarized in Table 3 show that NTP modulates immune cells via redox signaling consistent with the current hypothesis of NTP cellular action. Indeed, NTP-derived ROS and RNS can induce the immunological response in many cell types (for more information, see Table 3). In general, ROS and RNS are involved in many cellular processes, and at lower concentrations, they positively regulate the immune system [138]. Thus, the manipulation of ROS balance may be an interesting therapeutic approach in many diseases, including cancer [139]. Recently, a number of studies reported the activation of immunogenic cell death in cancer cells post NTP-treatment in vitro [128,129,140,141]. However, assessment solely enhanced levels of damage-associated molecular patterns (DAMPs, such as ATP increase, CRT activation) as a final proof of ICD in vitro is not sufficient. Thus, the potential of an agent to activate bona fide ICD has to be evaluated in vivo [142,143]. In fact, only a limited number of reports shows the ICD stimulation post direct or indirect (NTP-treated liquids) NTP treatment in vivo in appropriate animal models of oncogenesis [46,96].

However, in order to come closer to the real-life clinical approach, thorough in vivo validation should be performed. NTP shows some promising results in vivo as well. We summarized current in vivo studies on NTP-induced modulation of immune cell activity in Table 4. We have to say, that there are more in vivo studies dealing with NTP modulation of immune cell activity. However, the vast majority of those reports is rather descriptive in nature. Thus, it is not surprising that the molecular foundations for the alleged immunomodulatory effects remain generally enigmatic. In the absence of a hypothetical mechanism to guide experimental design, proper adjustment and control of the experimental parameters are usually precluded. Therefore, we selected in Table 4 studies that comply with following criteria: availability of statistical assessment, presence of positive controls for immunomodulatory assays, orthogonal validation of immunomodulation, and several replicates of proof of the concept experiments.

According to Table 4, the majority of in vivo studies utilized NTP-induced immune cell modulation in some kind of immunotherapeutic approach. Indeed, NTP technology has been shown to be effective in immunoprotection against malignant melanoma [46] or as a potential adjuvant melanoma treatment via induction of immunogenic cell death (ICD) [48]. Overall, this analysis of in vitro and in vivo studies on the immunomodulatory effects of NTP shows that NTP has the potential in mediating the activity of immune cells. Such modulation of immune cells functions by NTP shows the potential to effectively control tumor growth at least in a mouse model of melanoma [46]. However, the molecular mechanisms of NTP-induced immune cell modulation remain unclear. There is a need for future studies to elucidate this gap in knowledge.

## 5. Challenges in Deciphering Molecular Targets of NTP Action

Contrary to previous reviews on biomedical applications of NTP, we would like to add a bit of critical analysis here. In judging results and making straightforward conclusions, we have to be very careful. Biomedical literature faces rising concerns that a substantial fraction of published research findings are false [149,150]. Overall, modern science is hampered by the issue of reproducibility of the research [151,152]. Specifically, in many cases, biomedical studies take shortcuts around the used methodology, resulting in devastating consequences [153,154]. It is estimated, that low reproducibility rates within life science research result in approximately $28 billion USD/year being wasted on irreproducible preclinical research in the United States [155]. Thus, it is an imperative to critically assess potential treatment modalities.

Although NTP shows great potential, we have identified certain pitfalls in the current research, which create challenges in the identification of molecular mechanisms of NTP action. Importantly, in a long run strategy such challenges may result in the clinical fail of the treatment. Further, we briefly describe the major challenges in NTP-immune cell modulation. In Section 2, we summarized that NTP sources vary in used gas composition power, discharge voltage, gas flow rate, jet length, and voltage frequencies, which in turn dramatically affect the chemistry of NTP [24,30,33,45,54,55]. This variability in design and physicochemical composition of NTP greatly affect biological outcome modulated by NTP. In fact, NTP shows sometimes bewildering biological effects. As an example, from Table 3 and Table 4, one can see, that there is a huge variability in the type of device, gas composition, voltage and frequency. Bearing in mind how NTP-triggered biological effects vary, it is very difficult to compare isolated studies. Thus, there is a significant challenge in the standardization of NTP treatments. In order to reasonably compare different NTP effects in different laboratories, there is an unmet need for standardization of the treatment protocols. Just to illustrate how cautious one has to be, here is an example of how a very tiny handling protocol alteration may lead to irreproducible results [156]. Two laboratories could not reproduce each other’s cell-sorting profiles of breast cells, notwithstanding the fact that they utilized identical methods, reagents, and even specimens [156]. After long-term struggle, the researchers realized that the stirring procedure made a difference [156].

Another challenge lies in the usage of cell lines. From Table 3, it is apparent that the majority of studies, for instance, are done utilizing the THP-1 cell line. In fact, monocytic cell lines of varying degrees of differentiation represent a very nice initial model that can substitute primary innate immune cells, e.g., macrophages in vitro [157,158,159]. In order to closer mimic macrophages features, differentiation protocols using phorbol-12-myristate-13-acetate (PMA) or 1,25-dihydroxyvitamin D_3_ are frequently used. Indeed, such differentiation may recapitulate certain macrophage functions [160,161]. However, the phenotype of the differentiated cells is very different when compared with primary cells, reflecting differences in gene expression and altered cellular functions [160,162]. Thus, a broad involvement of human primary cells is crucial for deciphering mechanisms of NTP-induced immunomodulatory effects.

Further, despite the fact that trends in biomedical research are changing, to get US Food and Drug Administration (FDA) treatment approval, it is not necessary to identify the mechanism(s) of treatment action [52,53]. However, we should remember that such an approach might lead to severe failure at the final stages of clinical trials [52,53]. In fact, deciphering the mechanism of treatment action really matters and starts with target identification [52]. Importantly, target verification requires a thorough biological understanding [163]. As a result, target verification contributes greatly to a reduction in the rate of clinical failure of a treatment in early clinical development [163]. Knowledge of the mechanism by which a drug/treatment acts greatly helps to optimize the therapeutic window of a treatment [164]. When the mechanism of action is known, it is possible to perform better dosing for a patient via monitoring the drug’s effects on the target pathway [52,53,164]. Knowledge of how a drug/treatment works is essential to stratify clinical trials optimizing patient enrollment [52,53,164]. In case of developing a treatment that utilizes cytotoxic or cytostatic effects (e.g., anticancer or antibacterial), understanding the mechanisms of resistance and action at the molecular level is essential to develop a therapeutic modality capable of preventing or blocking resistance effect [165]. It is true that studies revealing molecular mechanisms of treatment action are costly, time-consuming, and require a lot of effort. After all, this knowledge pays off in the long run by increasing the chances for drug approval and saving money and time at the stage of clinical trials [52]. However, most importantly, it saves the lives of patients [52]. Therefore, it is very important to identify cellular and molecular mechanisms of NTP action. We have to admit that several studies have been undertaken (Table 3 and Table 4). However, there is a significant lack of research that utilizes gene-editing techniques to verify obtained findings. Indeed, the genetic background of cells may dramatically influence the susceptibility of cells to NTP treatment [37]. The cases of such research are unfortunately isolated. We have to grasp the necessary information from replicated robust studies to achieve effective NTP-based treatments. Thus, we definitely need more studies that reveal the underlying molecular mechanisms of NTP-induced immunomodulatory effects. Only the knowledge of the spatiotemporal mechanisms of the NTP-induced effects will enable the deliberate exploitation of such signals, e.g., for the potential clinical translation. Additionally, we need to realize our current misunderstandings on NTP-based treatments. This will put NTP in a better position to become a progressive treatment modality.

## 6. Conclusions

Concluding our review, we would like to emphasize that NTP really has great potential in various biomedical applications and particularly as an immunomodulatory effector. From the cellular biological point of view, the field is still in its infancy. The pitfalls that the NTP field faces are typical for developing research directions. Overall, in recent years, pharmaceutical drug research and development show declining output in terms of the number of new drugs [166]. However, as scientists, we must remember that the main goal of our research is to finally help patients by developing clinically useful treatments [167].

In this review, we summarized critical challenges that have to be addressed by the researchers in order to make NTP a reliable clinical treatment. We hope that our critical analysis will help researchers to overcome the aforementioned challenges and develop better controlled, safer, and more robust NTP-based treatment modalities.

## Figures and Tables

**Figure 1 ijms-21-06226-f001:**
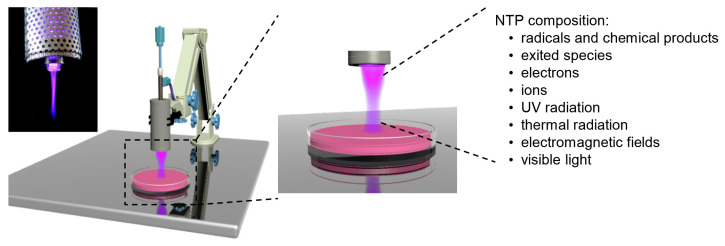
Schematics of atmospheric pressure plasma jet (APPJ) system with an image of the plasma torch. Image illustrates the entire physicochemical complexity of NTP.

**Figure 2 ijms-21-06226-f002:**
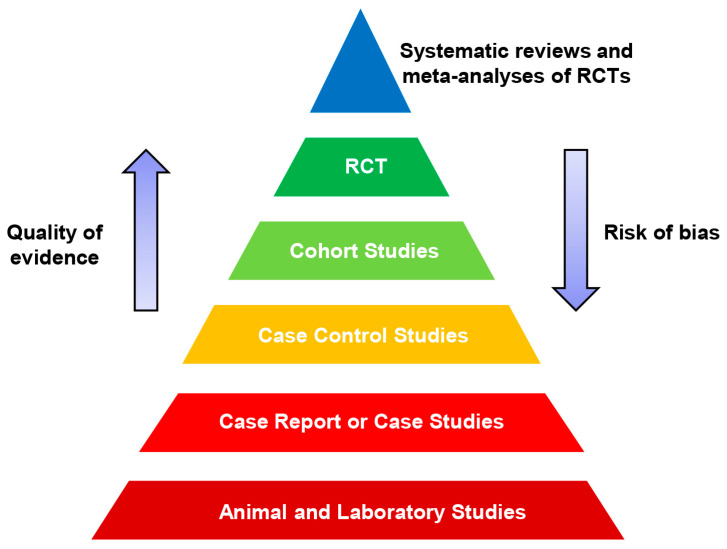
Hierarchy of evidence pyramid. RCT—randomized controlled trial.

**Table 1 ijms-21-06226-t001:** Major types of reactive oxygen and nitrogen species form by non-thermal plasma (NTP).

ROS or RNS Name	Chemical Formula
Superoxide anion	O_2_^−^
Hydrogen peroxide	H_2_O_2_
Hydroxyl radical	^•^OH
Singlet oxygen	^1^O_2_
Ozone	O_3_
Organic radicals	RO^•^, RO_2_^•^
Nitric oxide	^•^NO
Nitrogen dioxide	^•^NO_2_
Peroxynitrite	ONOO^−^

**Table 2 ijms-21-06226-t002:** Studies that assessed side effects of NTP.

Plasma Type	Pathological Condition	Side Effects	Type of Study	Ref.
APPJ	Chronic leg ulcers	No signs of cytotoxicity	Cohort study	[91]
DBD	Skin infection eczema	No side effects	Case study	[92]
APPJ	Chronic infected skin wounds	Pain (before and after treatment)	Clinical trial	[25]
APPJ	Skin ulcers	No side effects	Case study	[23]
APPJ	Head and neck cancer	Bad taste, pain, collateral edema, bleeding, sialorrhea, necrosis	Case control study	[78]
APPJ	Skin herpes zoster	No side effects	Clinical trial	[93]
APPJ	Skin psoriasis vulgaris	No side effects	Case study	[94]
APPJ	Skin chronic wounds	No side effects	Case control study	[95]
APPJ	Skin wounds	Focal mucosal erosion with superficial ulceration and necrosis accompanied by a mild inflammatory reaction.	Animal study	[79]

**Table 3 ijms-21-06226-t003:** Generalized summary of NTP modulation of immune cell activity in vitro.

Plasma Device	Physicochemical Parameters			Cell Lineage	Signaling Pathway	Main Results	Ref.
	Gas	Voltage (kV)	Frequency				
kiNPen 11	Ar	N.A.	~1 MHz	THP-1	Inflammation	↑*IL-8* mRNA level and secretion;	[124]
						↑*HMOX* mRNA level	
kINPen 09	Ar	2–6	~1 MHz	Jurkat and THP-1	Jurkat cells apoptosis,	↑resistance of THP-1 to plasma-treated medium in comparison to Jurkat cells;	[125]
					THP-1 anti-oxidant defense	differences in expression levels of genes involved in redox and anti-oxidant system regulation and apoptosis.	
APPJ	Air	2	N.A.	THP-1, U937 and RAW264.7, PBMCs	Apoptosis	Inhibition of cell growth;	[126]
						↓Glucose consumption,	
						intracellular ATP and lactic acid production;	
						mitochondria membrane depolarization,	
						cytochrome c release and induction of apoptosis.	
DBD	N_2_	1.08	30 kHz	T98G and A549 in co-culture	Macrophage activation, cancer cells death induction	↑expression of *iNOS* and *TNF-α* genes on mRNA and protein levels;	[121]
				with RAW264.7		plasma-activated macrophages induced the cell death of glioma and adenocarcinoma in co-culture	
kINPen 11	Ar	N.A.	1 MHz	Neutrophils isolated from	NETosis	Activation of NETosis in neutrophils; Release of DNA, extracellular DNA.	[127]
				venous blood			
kINPen	Ar	N.A.	N.A.	THP-1, A375, primary monocytes	Alternation in metabolic activity	Altered the morphology of THP1 cells; changes in surface markers expression; ↑IL8 and MCP-1 in PMA-stimulated THP-1 ↑ IL1β, IL6, and IL8	[122]
				isolated from PBMCs	and morphology	↑HLA-DR (an M1 macrophage marker) and fibronectin (and M2 macrophage marker)	
DBD	Air	29	15 and 30 Hz	THP-1, A549 in co-culture	ICD	induction of ICD in A549 cells	[128]
						(↑calreticulin, ROS production, ATP secretion);	
						↓viability of Plasma treated A549 cells,	
						when co-cultured with M0 macrophages	
DBD	N.A.	29	5, 15, 30, 75 Hz	CNE-1, THP-1	ER stress, ICD	↑ immunogenic cell death of cancer cells;	[129]
						↑ATP secretion;	
						↑ER stress proteins (↑ATF4-STC2 pathway).	
kiNPen	Ar	2–6	1.1 MHz	Jurkat, THP-1	Apoptosis	↑resistance of THP-1 cells to plasma treatment in comparison with Jurkat cells,	[130]
						↑ caspase 3 dependent apoptosis;	
						↑ERK 1/2 and MEK 1/2 and p38 MAPK and JNK 1/2;	
						↑HSP27 in THP-1.t	
kiNPen	Ar	N.A.	1 MHz	Jurkat, U-937	Apoptosis, Ferroptosis	Plasma treatment in combination with pulsed electric fields (electro square porator)	[19]
						resulted in ↑cytotoxicity in Jurkat cells. Contrary, the additive effect was smaller in U937 cells;	
						activation of apoptosis;	
						↑ROS production, caspase 3/7 activation).	
DBD	Air	20	500 Hz	Jurkat	Apoptosis	↑p53 protein, but not on mRNA level 48 h post plasma treatment;	[131]
						↑Bax and Bcl-2 proteins after 24 h, slightly ↑caspase-8;	
						↑mRNA levels of antioxidant enzyme *SOD1*, *CAT*, and *GSR2* 6 and 24 h post NTP treatment	
						as a response to ROS elevated oxidative stress	
kINPen	Ar	N.A.	N.A.	TK6	DNA damage response	↑γH2AX post plasma treatment as a consequence of ROS induced	[132]
						oxidative stress in apoptosis	
DBD	Air	25	20 kHz	Human monocytes isolated from venous blood, MDM	ROS production, surface markers expression	↓CD86, CD36, CD163 and CD206;	[133]
						↓CD16 post NTP treatment;	
						NTP treatment of MDM led to time-dependent ↓M1 population, significantly after 30 sec of treatment, following ↑M2 population.	
kINPen MED	Ar	N.A.	N.A.	MBMDc, PDA6606 in co-culture	Macrophage polarization	↑NOS2 in TAM;	[123]
						slight ↑M2 polarized macrophages post exposure with plasma- treated medium;	
						↑CXCL1 and CCL4 in non-polarized macrophages post plasma-treated medium;	
						↓CXCL1, CCL4, MCP1 in TAM.	
kINPen	Ar	2–6	1 MHz	splenocytes of mice spleens, B16F10 in co-culture	Immune cells activation	↓metabolic activity in naive and PMA-stimulated splenocytes;	[134]
						↑IL-10, CCL4, IL-4, IL-12, and IL-1β in naive splenocytes;	
						↑calcium influx in splenocytic T-cells, but not in macrophages;	
						Co-culturing of monocytes with plasma-treated melanoma cells ↑CD115, IL-10 and CCL4, with a slightly ↑IL-1β, IL-12p70, TNFα, and TGFβ.	
						Co-culture of CD4+ T helper and CD8+ cytotoxic T cells with plasma-treated melanoma cells showed an increase of CD4 over CD8 cells (↑CD28).	

ATP—adenosine triphosphate; ATF4—activating transcription factor 4; CAT—catalase; CCL4—carbon tetrachloride; CXCL1—C-X-C motif ligand 1; ER—endoplasmic reticulum; HLA-ABC—human leukocyte antigen ABC; HMOX—heme oxygenase; HSP27—heat shock protein 27; ICD—immunogenic cell death; iNOS—nitric oxide synthase gene; MBMDc—murine bone-marrow derived cells; MCP1—monocyte chemoattractant protein; MDM—monocyte-derived macrophages; N.A.—not assessed; NET—neutrophil extracellular traps; NETosis—neutrophil extracellular traps activation and release; NOS2—nitric oxide synthase; PBMCs—peripheral blood mononuclear cells; PMA—phorbol-12-myristate-13-acetate; SOD1—superoxide dismutase 1; STC2—stanniocalcin-2; TAM—tumor-associated macrophages; TGFβ—transforming growth factor beta; TNFα—tumor necrosis factor alpha; VBN—venous blood neutrophils; Z-VAD-FMK—carbobenzoxy-valyl-alanyl-aspartyl-[O-methyl]- fluoromethylketone; ↑—upregulation; ↓—downregulation.

**Table 4 ijms-21-06226-t004:** Summary of NTP modulation of immune cell activity in vivo.

Plasma Device	Physicochemical Parameters			Animal Model	Signaling Pathway	Main Results	Ref.
	Gas	Voltage (kV)	Frequency				
kINPen MED	Ar	N.A.	N.A.	C57BL/6 mice	Immuno-modulation	↓total number of tumor nodes;	[144]
						↑infiltration of macrophages, but not CD206+ cells into tumors;	
						↑ number of macrophages and T cells,	
						with no changes in numbers of dendritic cells and neutrophils. Increased level of calreticulin	
kINPen MED	Ar	N.A.	1 MHz	C57BL/6 mice	Apoptosis in tumor tissue	Induction of apoptosis in tumor tissues;	[145]
						No significant differences in the number of granulocytes, monocytes, and lymphocytes in general;	
						No changes in cytokines secretion of IL6, IL10, IL12, MCP1, IFNγ, or TNFα.	
APPJ	O_2_ or N_2_	24	N.A.	CD2F1 and C57BL/6 mice	Tumor growth inhibition	↓tumor size in CD2F1 mice;	[146]
						↑IFN-γ, no changes in TNF-α from splenocytes of the plasma-treated CD2F1 mice;	
						In the C57BL/6 mice very weak response to plasma-treatment;	
						Discussion on immune response, but no data are provide to	
						support it.	
kINPen	Ar	N.A.	N.A.	Balb/C mice	ICD	↑immunogenic cell death markers in CT-26 cells;	[96]
						heat shock protein 70 (HSP70), and high-mobility-group-protein B1 (HMGB1);	
						↑IL1β, IL6, IL12p70, CCL4, and TNFα.	
						↑number of macrophages and T cells in mice	
						with CT26 peritoneal carcinomatosis post treatment with oxidized saline solution.	
DBD	Air	17	50–500 Hz	C57BL/6J mice	ICD	Activation of immunogenic cell death marker (calreticulin);	[48]
						↑survival rate of mice post vaccine injection prepared from B16F10 melanoma cells treated with DBD plasma.	
kINPen	Ar, Ar+O_2_, He, He+O_2_	N.A.	1 MHz	C57BL/6 mice	ICD	↓tumor growth	[46]
						↑CD8+ cytotoxic T-cells;	
						↑macrophages;	
						↑CD11c+ dendritic cells (DCs);	
						↑CD127 in both CD4+ and CD8+ T-cells;	
						↑ICD markers in B16F10 (↑CRT, HSP90, CD47);	
						Co-culture of splenocytes isolated from vaccinated mice with B16F10 ↑marker CD69 in CD8+ T cells and ↑CXCL1, CXCL10, IFNγ, IL1α, IL6, and TNFα;	
						↓GM-CSF, CCL17.	
APPJ	N_2_	N.A.	N.A.	C57/BL6 mice	Anti-inflammatory effect	↓immune cells infiltration (CD4+ T cells, CD11c+ cells, CD11b+ cells, and Gr-1+ cells);	[147]
						↓pro-inflammatory cytokine and chemokine (IL-6, IL-17, IL-22, CCL20 and CXCL1);	
						↓Th17 cell differentiation in lymph node;	
						In vitro suppressed differentiation of naive CD4+T cells into Th17 cells and Th1 cells;	
						↓CD80, CD86, and MHCII in BDCM and ↓*IL-6* expression *TNF-α* and *IL-6*.	
APPJ	N_2_	5	15 kHz	NC/Nga mice	Anti-inflammatory effect	In vivo: NTP treatment ↓HDM-induced infiltration of mast cells and eosinophil into the dermis and ↓Th2 cell differentiation;	[148]
						↓TSLP and CCL17 post NTP treatment in HDM-induced AD;	
						In vitro: Activated mast cells incubation in plasma- treated medium resulted in ↓*NF-κB*, *TNF-α*, *IL-6* and *IL-13*	

AD—atopic dermatitis; BMDC—bone marrow-derived dendritic cells; CCL17—chemokine (C-C motif) ligand 17; CRT—calreticulin; CXCL1—C-X-C motif ligand; DCs—dendritic cells; GM-CSF—granulocyte-macrophage colony-stimulating factor; HDM—house dust mite; HMGB1—high mobility group protein B1; HSP70—heat shock protein 70; IFNγ—interferon gamma; IL—interleukin; MHC II—major histocompatibility complex class II; N.A.—not assessed; NF-κB—nuclear factor kappa B; NK cells—natural killer cells; PBS—phosphate-buffered saline; TNFα—tumor necrosis factor alpha; TSLP—thymic stromal lymphopoietin; ↑—upregulation; ↓—downregulation.

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
