# Peer review of "Critical Analysis of Non-Thermal Plasma-Driven Modulation of Immune Cells from Clinical Perspective"

_ijms, 2020, doi:10.3390/ijms21176226_

Round 1

Reviewer 1 Report

The article "Analysis of Non-thermal Plasma-Driven Modulation of Immune Cells" is an attempted review of the published work on this topic. The topic is timely but the manuscript needs significant overhaul for it to be publication-ready. Here are some of my suggestions for improvement:

  1. Please use the help of a language editor.
  2. There is a lack of focus. The manuscript bounces between plasma, cellular effects and clinical practice and this reviewer is unable to decipher what the message is. It certainly does not discuss the topic in depth.
  3. The sections in the manuscript need some reorganization. As an example, Physicochemical characterization and side effects have no relationship. One is a lab-based evaluation, the other is patient-based - unless there is significant literature that shows a direct cause-effect relationship.
  4. Table 2: The logic behind this table is not clear. What is the message? Clearly, the side effects are not device related. Is the toxicity related to disease? Or to site of delivery? Or duration and frequency of treatment? I think this whole section is irrelevant and should be deleted.
  5. Figure 2: Is irrelevant to this topic. I am not sure what CPGs have to do with the topic of this review.
  6. Table 3: One again, I am not sure what message the authors are trying to convey. The organization of the table adds to the confusion. No distinction is made between research on direct plasma exposure of  immune cells versus immune function alteration as a secondary effect of plasma exposure of diseased cells. Human versus animal? Organized by device? Organized by disease? Organized by cell lines vs. primary cells.
  7. Table 4: Many studies are missing from this summary table. Summary usually does not exclude studies based on some random criteria. It is supposed to be all-encompassing. I am not sure you will make many friends if you exclude people's work and make the statement "We have to grasp the necessary information from replicated robust studies to achieve effective NTP-based treatments"
  8. Please perform a thorough analysis of the literature that is relevant to your title. While there is extensive review of other literature, I feel the most emphasis should be placed on research that is directly related to this topic. Otherwise only a partial (maybe skewed) analysis is presented. Or, I suggest change the title.
  9. Question of treatment standardization and comparison of studies: I will use the example of a gas oven versus an electric oven. Different approaches for baking a cake - all you need to know is if the cake baked!! So, if there is efficacy, no one cares about the mode of action. Paracetamol has been used for decades now and we don't know it's mechanism of action!!
  10. Minor issues: please check your facts. THP is a monocytic lineage cell line. Jurkats are T cells. You talk about inhibition of angiogenesis, when there are several studies that show plasma-stimulated angiogenesis. After all, wound healing cannot proceed without angiogenesis! There are other such misrepresentations.
  11. If you are talking about modulation of immune cells, leave clinical application, FDA approval etc. out of this review.

Author Response

POINT BY POINT REPLY TO THE REVIEWER’S COMMENTS

We would like to thank the Reviewer for his/her review and criticism that helped us to improve the quality of our paper. In fact, the Reviewer’s comments confirmed necessity and importance of points and especially challenges in the field of plasma medicine raised in our manuscript.

Detailed point-by-point responses to all Reviewer’s remarks together with the corresponding amendments made to the manuscript are provided below.

Reviewer’s questions and comments are given in italic; replies are given in blue and the changes in the manuscript are marked in red.

Reviewer #1 (Comments to the author):

  1. Please use the help of a language editor.

The manuscript has been checked by a native speaker. 

  1. There is a lack of focus. The manuscript bounces between plasma, cellular effects and clinical practice and this reviewer is unable to decipher what the message is. It certainly does not discuss the topic in depth.

The focus of the manuscript is to provide a critical review on plasma modulation of immune cell functions from clinical perspective. The manuscript appeals to broad readership raging from physicists to clinicians. For that reason, it is crucial to at least briefly describe what is plasma and its physicochemical characteristics. Next, as far as, there are substantial efforts to put plasma into clinical routine, it is crucial to add current view on validity and efficacy of clinical research trials and translational research. In order to propel any novel potential treatment to successful clinical application, it is very important to critically assess current literature, specifically identifying weak spots, challenges and inconsistencies in the literature.

In order to avoid any further misunderstanding, we clarified our concept in the revised version. 

  1. The sections in the manuscript need some reorganization. As an example, Physicochemical characterization and side effects have no relationship. One is a lab-based evaluation, the other is patient-based - unless there is significant literature that shows a direct cause-effect relationship.

We re-organized manuscript accordingly and added appropriate clarifications.

  1. Table 2: The logic behind this table is not clear. What is the message? Clearly, the side effects are not device related. Is the toxicity related to disease? Or to site of delivery? Or duration and frequency of treatment? I think this whole section is irrelevant and should be deleted.

We are sorry that the Reviewer did not get the idea behind the Table 2. Indeed, the message is to illustrate how frequently the question of plasma’s side effects is omitted in the literature.

Concerning Reviewer’s commentClearly, the side effects are not device related.” There is only one study with DBD plasma and rest are with APPJ. Thus, one cannot make any reasonable conclusion that the side effects are not device related. Parameters of treatments vary as well. In addition, sampling size of the studies is very low. Thus, it is hard to reasonably evaluate side effects. That was exactly our endeavor. We wanted to show how little we know about NTP’s side effects.

In order to avoid any further misunderstanding, we clarified our concept and added pathological conditions to the Table 2 in the revised version.

  1. Figure 2: Is irrelevant to this topic. I am not sure what CPGs have to do with the topic of this review.

We would like to stress once again that this is a critical review from clinical perspective. As far as NTP treatment endeavors to attain different clinical applications, NTP field has to take a sip of current view in CPGs. It is important to show in such critical review current progress in evidence-based medicine. It is important to discuss, that the views on medical research has changed. Now we recognize limitations of evidence alone, and increasingly stress the need to combine critical appraisal of the evidence. Evidence-based research has shaped clinical medicine by placing the practice of medicine on a solid scientific basis and developing more sophisticated hierarchies of evidence. These things need to be at least briefly mentioned in the light of critical analysis of NTP effects desired to be a clinically relevant treatment in the future.

In order to avoid any further misunderstanding, we clarified our concept in the revised version.

  1. Table 3: One again, I am not sure what message the authors are trying to convey. The organization of the table adds to the confusion. No distinction is made between research on direct plasma exposure of immune cells versus immune function alteration as a secondary effect of plasma exposure of diseased cells. Human versus animal? Organized by device? Organized by disease? Organized by cell lines vs. primary cells.

In fact, Table 3 shows generalized summary of NTP modulation of immune cell activity in vitro. There is no need, yet, to do any distinction between research on direct plasma exposure of immune cells versus immune function alteration as a secondary effect. Because, in general, there is very little knowledge about molecular mechanisms of plasma action on immune cells. The whole manuscript is about it. We want to stress that there is an unmet need to fill this gap.

Indeed, the Reviewer distorts and shuffles the facts. The Table 3 organization is straightforward. There is a summary of devices with their characterization; then there are cell types which were subjected to NTP; then – signaling that was affected; and main results, which reflect immune cell functionality. This is exactly the type of a table which one would expect in the frame of critical review. The questionsHuman versus animal? Organized by device? Organized by disease? Organized by cell lines vs. primary cells.are completely irrelevant in this context. First of all, mice, for example, are not men, especially in terms of inflammation models [1]. Thus, it would be very spurious to compare inflammatory effects in human versus animal. Second, characterization of NTP and device type is highlighted in the Table. Third, we present a generalized summary, thus there is no need to indicate disease. Moreover, it is very premature to organize such summary in accordance with disease, because there is substantial lack of information concerning NTP-induced immune cell modulation. This is exactly the aim of the Table 3. Indeed, we honestly highlight which cell types were used in each study. Further, in the part “Challenges in Deciphering Molecular Targets of NTP Action” we discuss why primary cells matter for the clinical translational research. Thus, the Reviewer’s commentOrganized by cell lines vs. primary cells.is absolutely irrelevant. Once again, we intended to emphasize, that there is really a lack of knowledge concerning elicited immunomodulatory effects of NTP. 

  1. Table 4: Many studies are missing from this summary table. Summary usually does not exclude studies based on some random criteria. It is supposed to be all-encompassing. I am not sure you will make many friends if you exclude people's work and make the statement "We have to grasp the necessary information from replicated robust studies to achieve effective NTP-based treatments"

Indeed, we present a critical review here. We thought, that we do not need to state this in the title of the manuscript. We apologize for this oversight. A critical review should not be all-encompassing. The purpose of critical literature assessment is to give a reader balanced view on current research topic highlighting both positive and negative results. Therefore, not all papers should be included in the review. We elaborated in the revised manuscript criteria for the selection of studies. Furthermore, we extended explanation for the necessity of critical reviews in the Introduction and in the part “Challenges in Deciphering Molecular Targets of NTP Action”.

The Reviewer misinterpreted our phrase "We have to grasp the necessary information from replicated robust studies to achieve effective NTP-based treatments". We wrote it to underline, that more research, indeed, is needed. In order to avoid any further misunderstanding, we clarified this question in the revised version.

We are really confused by the Reviewer’s statementI am not sure you will make many friends if you exclude people's work and make the statement "We have to grasp the necessary information from replicated robust studies to achieve effective NTP-based treatments"”. Because such statements are against the COPE Ethical Guidelines for Peer Reviewers [2].

  1. Please perform a thorough analysis of the literature that is relevant to your title. While there is extensive review of other literature, I feel the most emphasis should be placed on research that is directly related to this topic. Otherwise only a partial (maybe skewed) analysis is presented. Or, I suggest change the title.

We did a thorough analysis of the literature. We included criteria for the selection of papers. We elaborated repeatedly why critical analysis is essential. As suggested, we changed the title of the revised manuscript. 

  1. Question of treatment standardization and comparison of studies: I will use the example of a gas oven versus an electric oven. Different approaches for baking a cake - all you need to know is if the cake baked!! So, if there is efficacy, no one cares about the mode of action. Paracetamol has been used for decades now and we don't know it's mechanism of action!!

We are thankful to the Reviewer for that comment. In fact, final cake quality very much depends on airflow, oven temperature profiles, the internal cake temperature and physical parameters of oven [3]. Thus, it really matters what kind of oven one uses to bake a cake.

Reviewer’s statement, that no one cares about mechanisms of action, contradicts with modern pharmacological science [4]. Understanding the mechanisms of drug/treatment action at the molecular level has far going implications. It helps to optimize the therapeutic window of a treatment; enables better dosing for a patient; stratifies clinical trials; saves the lives of patients. “Saving lives of patients” is not simply words. There is a classic example of thalidomide, where lack of knowledge of molecular mechanisms of action led to a tragedy [5]. We elaborated this part in the revised version of the manuscript.

We are very delighted that the Reviewer used paracetamol example. However, the Reviewer distorts and shuffles facts and definitions. By strictly statingwe don't know it's mechanism of action!!’ the Reviewer probably points towards 100% “proof” which indicates that there is no room for error. Frankly speaking, if we take any scientifically established process, we will find that we do not know “precisely enough” any mechanism of action. There will be always a room to improve and polish our scientific view, because scientific research is the endless frontier [6]. There is a big variability and substantial difference in the validity of how one is imprecise about natural phenomena [7]. Fortunately, we have scientific method that helps us to select those models that more precisely, reliably and rigorously describe natural phenomena and give reliable outcomes or correct predictions [8]. In fact, paracetamol’s mechanism of action is yet to be fully determined. However, we know a lot about its mechanism of action. There are established efficacy, dosages, clinical outcomes, pharmacokinetics, pharmacodynamics, toxicity, excretion, molecular and cellular targets of its action [9-16]. There is a systematic review on side effects of paracetamol [17]. There are systematic reviews and meta-analysis of its efficacy, for example [18, 19]. The situation in plasma medicine field is very different. That is why we need critically analyze our current knowledge, identify gaps and challenges. For the successful clinical translation, we have to understand the limitations of the technology. Importantly, we have to identify the misconceptions pervasive in the field, because only then we would be able to propel the field further. Acceptance of the failure, with making reasonable conclusion of it, is a road to success [20].

  1. Minor issues: please check your facts. THP is a monocytic lineage cell line. Jurkats are T cells. You talk about inhibition of angiogenesis, when there are several studies that show plasma-stimulated angiogenesis. After all, wound healing cannot proceed without angiogenesis! There are other such misrepresentations.

The Reviewer is not correct sayingplease check your facts. THP is a monocytic lineage cell line. Jurkats are T cells”. We never stated in the text the opposite. We wrote the following passage: “THP-1 cell lines showed greater resistance to plasma treatment in comparison to primary monocytes [128] and Jurkat cells [116,121]. NTP-treatment led to p53 [122] and caspase3/7 [121] activation and apoptosis execution in Jurkat cells.” Where we compared NTP effects in different immune-like cell types, namely THP-1, primary monocytes (this is not THP-1 or Jurkat) and Jurkat cells. In order to avoid any further misunderstanding, we added specification of cell lines in the revised version.

Talking about angiogenesis, the Reviewer again distorts and shuffles the facts. We did not discuss or focus on angiogenesis inhibition, especially in relation to wound healing, which is not a topic of current review either. In the original manuscript ‘angiogenesis suppression’ was mentioned only once in the introduction to illustrate variability of NTP applications: “Indeed, non-thermal plasma (NTP) has emerged as a promising tool for sterilization of medical equipment [6-8], wound healing [9,10], bacteria eradication [11-14], dental hygiene [6,15], blood coagulation [16], angiogenesis suppression [17], cancer treatment [4,18,19] and food decontamination [20,21].” However, we added in the revised version sentence about studies that show angiogenesis stimulation. 

  1. If you are talking about modulation of immune cells, leave clinical application, FDA approval etc. out of this review.

We would like to emphasize once again, that this is a critical review from a clinical perspective. NTP is meant to achieve clinical translation. Thus, at least brief knowledge of FDA approval, clinical trials and Clinical Practice Guidelines is absolutely necessary in order to reach that. Therefore, we deliberately included those descriptions. This helps to understand the demands of clinical translation and to orient where currently NTP applications stay. We elaborated this amendment in the revised version of the manuscript.

References:

  1. Seok, J., et al., Genomic responses in mouse models poorly mimic human inflammatory diseases. Proc. Natl. Acad. Sci. U. S. A., 2013. 110: p. 3507-12.
  2. https://publicationethics.org/resources/guidelines-new/cope-ethical-guidelines-peer-reviewers.
  3. Shahapuzi, N.S., et al., Effect of oven temperature profile and different baking conditions on final cake quality. International Journal of Food Science and Technology, 2015. 50: p. 723-729.
  4. Brunton, L.L., B.r.C. Knollmann, and R. Hilal-Dandan, Goodman & Gilman's : the pharmacological basis of therapeutics. 2018.
  5. Kim, J.H. and A.R. Scialli, Thalidomide: the tragedy of birth defects and the effective treatment of disease. Toxicol Sci, 2011. 122: p. 1-6.
  6. Park, K., Science the endless frontier. J. Control. Release, 2019. 308: p. 240.
  7. Asimov, I., The relativity of wrong. The Skeptical Inquirer, 1989. 14: p. 35-44.
  8. Radnitzky, G. and W.W. Bartley, Evolutionary epistemology, rationality, and the sociology of knowledge. 1988, Open Court: La Salle, Ill.
  9. Sharma, C.V. and V. Mehta, Paracetamol: mechanisms and updates. Continuing Education in Anaesthesia Critical Care & Pain, 2013. 14: p. 153-158.
  10. Botting, R. and S.S. Ayoub, COX-3 and the mechanism of action of paracetamol/acetaminophen. Prostaglandins Leukot Essent Fatty Acids, 2005. 72: p. 85-7.
  11. Graham, G.G., et al., The modern pharmacology of paracetamol: therapeutic actions, mechanism of action, metabolism, toxicity and recent pharmacological findings. Inflammopharmacology, 2013. 21: p. 201-32.
  12. Graham, G.G. and K.F. Scott, Mechanism of action of paracetamol. Am J Ther, 2005. 12: p. 46-55.
  13. Jozwiak-Bebenista, M. and J.Z. Nowak, Paracetamol: mechanism of action, applications and safety concern. Acta Pol Pharm, 2014. 71: p. 11-23.
  14. Anderson, B.J., Paracetamol (Acetaminophen): mechanisms of action. Paediatr Anaesth, 2008. 18: p. 915-21.
  15. Brandt, K.D., S.A. Mazzuca, and K.A. Buckwalter, Acetaminophen, like conventional NSAIDs, may reduce synovitis in osteoarthritic knees. Rheumatology, 2006. 45: p. 1389-1394.
  16. Hogestatt, E.D., et al., Conversion of acetaminophen to the bioactive N-acylphenolamine AM404 via fatty acid amide hydrolase-dependent arachidonic acid conjugation in the nervous system. J Biol Chem, 2005. 280: p. 31405-12.
  17. Roberts, E., et al., Paracetamol: not as safe as we thought? A systematic literature review of observational studies. Annals of the Rheumatic Diseases, 2016. 75: p. 552-559.
  18. Hyllested, M., et al., Comparative effect of paracetamol, NSAIDs or their combination in postoperative pain management: a qualitative review. British Journal of Anaesthesia, 2002. 88: p. 199-214.
  19. Wiffen, P.J., et al., Oral paracetamol (acetaminophen) for cancer pain. Cochrane Database of Systematic Reviews, 2017.
  20. Firestein, S., Failure : why science is so successful. 2016.

Reviewer 2 Report

Dear Authors,

This is a great review explaining the effects of non-thermal plasma in immunomodulation and its potential applications to biomedicine, as well as its challenges. The review is well-organized covered all these aspects. I only have one suggestion related to the tables. On “main results” is it possible being more concise? The tables will be more clear, and if the readers would like to explore more in deep, they can always go to the reference.

Thank you very much.

Author Response

POINT BY POINT REPLY TO THE REVIEWER’S COMMENTS

We would like to thank the Reviewer for his/her careful and rigorous review and constructive criticism that helped us to improve the quality of our paper.

Detailed point-by-point responses to all Reviewer’s remarks together with the corresponding amendments made to the manuscript are provided below.

Reviewer’s questions and comments are given in italic; replies are given in blue and the changes in the manuscript are marked in red.

Reviewer #2 (Comments to the author):

Dear Authors,

This is a great review explaining the effects of non-thermal plasma in immunomodulation and its potential applications to biomedicine, as well as its challenges. The review is well-organized covered all these aspects. I only have one suggestion related to the tables. On “main results” is it possible being more concise? The tables will be more clear, and if the readers would like to explore more in deep, they can always go to the reference.

Thank you very much.

We are thankful the Reviewer for these favorable comments. As requested we improved Tables 3 and 4.